



# Evaluating an exponential respiration model to alternative models for soil respiration components in a Canadian wildfire chronosequence (FireResp, v1.0)

John Zobitz[1, *], Heidi Aaltonen[2, *], Xuan Zhou[3, *], Frank Beninger[3, *], Jukka Pumpanen[2, *], and Kajar Köster[4, *]

[1]Augsburg University, Minneapolis, Minnesota, United States
[2]Department of Environmental and Biological Sciences, University of Eastern Finland, Kuopio, Finland
[3]Department of Environmental and Biological Sciences, University of Eastern Finland, Joensuu, Finland
[4]Department of Forest Sciences, University of Helsinki, Helsinki, Finland
[*]These authors contributed equally to this work.

**Correspondence:** Kajar Köster (kajar.koster@helsinki.fi)

**Abstract.** Forest fires modify soil organic carbon and suppress soil respiration for many decades since the initial disturbance. The associated changes in soil autotrophic and heterotrophic respiration from the time of the forest fire however, is less well characterized. We analyzed models of soil autotrophic and heterotrophic respiration with a novel dataset across a fire chronosequence in the Yukon and Northwest Territories of Canada. The dataset consisted of soil incubation experiments and field measurements of soil respiration and soil carbon stocks. The models ranged from a $Q_{10}$ (exponential) model of respiration to models of heterotrophic respiration using Michaelis-Menten kinetics parameterized with soil microbe carbon. For model evaluation we applied model selection metrics (Akaike Information Criterion) and compared predicted patterns in soil respiration components across the chronosequence. Parameters estimated with data from the 5 cm soil depth had better model-data comparisons than parameters estimated with data from the 10 cm soil depth. The model-data fit was improved by including parameters estimated from soil incubation experiments. Models that incorporated microbial carbon with Michaelis-Menten kinetics reproduced patterns in autotrophic and heterotrophic soil respiration components across the chronosequence. Autotrophic respiration was associated with aboveground tree biomass at more recently burned sites, but this association was less robust at older sites in the chronosequence. Our results provide support for more structured soil respiration models than standard $Q_{10}$ exponential models.

## 1 Introduction

While containing 15% of the total global soil area, high-latitude permafrost soils contain a significant proportion of global organic matter as well as global soil carbon content (Schuur et al., 2008; McGuire et al., 2009). These high-latitude regions are warming faster than the rest of the world, consequentially leading to (1) drier soils during the spring and summer (Masrur et al., 2018), (2) increases in the intensity and frequency of forest fires (Walsh et al., 2020), and (3) destabilization of the permafrost extent (Schuur et al., 2008, and McGuire et al. (2009)). For these regions, the combination of the above factors may lead to





increased release of soil $CO_2$ into the atmosphere from soil organic matter (Abbott et al., 2016). Soil respiration represents the product of several semi-independent processes: autotrophic (root) respiration (denoted here as $R_A$) and heterotrophic respiration (denoted here as $R_H$). Heterotrophic respiration consists of microbial respiration of labile carbon and microbial respiration associated with the breakdown of dead organic matter and other byproducts (Bosatta and Ågren, 2002; Harmon

et al., 2011). Autotrophic and heterotrophic respiration will also be affected by permafrost warming: while $R_A$ is strongly associated with primary productivity (Vargas et al., 2010; Pumpanen et al., 2015), $R_H$ may increase due to priming by newly accessible soil substrate (Fan et al., 2013; Karhu et al., 2016).

In high-latitude forests, soil respiration fluxes and soil carbon stocks exhibit variation depending on the time since the last wildfire (Bond-Lamberty et al., 2004; O'Donnell et al., 2011). Fire modifies soil organic carbon quality, making it harder for

microbes to access carbon (Holden et al., 2016; Song et al., 2019; Zhao et al., 2021). A recent meta-analysis by Ribeiro-Kumara et al. (2020b) of 32 studies measuring soil respiration following wildfires indicates two emergent patterns. First, overall soil respiration stabilizes 10-30 years following a fire. Second, for components of soil respiration, $R_A$ will increase and ultimately approach a steady-state value associated with forest succession and vegetation regrowth. On the other hand $R_H$ may decrease by association with post-fire changes in soil organic matter quality, temperature, or moisture (Aaltonen et al., 2019a, b; Wei

et al., 2010). For a sense of the magnitude of these changes, Bond-Lamberty et al. (2004) found the proportion of annual soil respiration that is $R_A$ changes from 5% (following disturbance), to 40% (21 years post-disturbance), and returning to 15% (150 years post-disturbance). The robustness of any patterns in $R_A$ and $R_H$ is highly uncertain given known soil heterogeneity in these high-latitude soils (e.g. permafrost versus non-permafrost soils, microbial versus fungal species composition).

Observations of overall soil respiration can be linked with process-based soil models to estimate $R_A$ and $R_H$. Models can

span a range from empirical models (Köster et al., 2017) to highly structured models of interacting soil microbes (Allison, 2014; Allison et al., 2018). There is agreement that a more detailed structural representation of microbial processes is needed in ecosystem models (Shao et al., 2013; Wieder et al., 2013, 2015; Luo et al., 2016; Vereecken et al., 2016). Improving the structural representation of microbial respiration in earth system models (e.g. accounting for microbial acclimation to non-equilibrium temperature changes, Zobitz et al. (2008); Wieder et al. (2013); Wang et al. (2021)), when appropriately

benchmarked with data, may reduce uncertainties in the turnover and stabilization of soil carbon (Wieder et al., 2013; Sihi et al., 2016). However, there are two main challenges to developing and evaluating more complicated soil process models. First, soil incubation studies may lead to underestimation of soil respiration components at larger scales (Reichstein and Beer, 2008; Hamdi et al., 2013; Chakrawal et al., 2020; Jian et al., 2020). Second, more complex models may lead to model equifinality - or when different models yield similar results (Tang and Zhuang, 2008). The combination of these multiple factors poses

challenges to both systematically develop and evaluate different soil respiration models. The objective of many modeling activities (especially for the remote sites studied here) is to strike a balance between modeling complex processes (Burnham and Anderson, 2002) while also parameterizing a model with available site measurements.

We have previously measured soil biogeochemical properties (stocks and associated respiration rates) across an established fire chronosequence in the Yukon and Northwest Territories in Canada (Köster et al., 2017; Aaltonen et al., 2019a, b; Zhou

et al., 2019). Our previous work focused on empirical associations between respiration and biogeochemical and environmental





measurements (e.g. soil organic matter, microbial content, and temperature) across the fire chronosequence. These results included both field measurements and soil incubation studies. For this study we synthesize both types of measurements across the chronosequence to parameterize process models of $R_A$ and $R_H$ (German et al., 2012; Todd-Brown et al., 2012; Sihi et al., 2016). We investigate two specific hypotheses in this study:

1. The association of autotrophic respiration with the time since disturbance is caused by an underlying positive association of $R_A$ with foliage biomass.

2. When corroborated with observational data, soil models that incorporate microbial carbon will better replicate the observed dynamics and associated fluxes ($R_A$, $R_H$, and the ratio $R_A/R_S$) across the fire chronosequence.

To evaluate our hypotheses we combine data from soil incubation experiments (Aaltonen et al., 2019b) with field data
(Köster et al., 2017) at chronosequence sites. For both incubation and field data, measurements were collected at the same time from similar plots to minimize any spatial and temporal biases in the data. Models are evaluated based on their ability to replicate measured soil respiration (both from incubation and field measurements). To reduce any biases with model fitting or model equifinality (Christiansen, 2018; Marschmann et al., 2019) we evaluate a range of parameter estimation approaches and data types.

## 2 Methods

### 2.1 Study sites

In 2015 we established a transect of sites in the northern boreal forests of Canada (Figure 1). All of these sites are located near Eagle Plains, Yukon (66° 220' N, 136° 430' W), and Tsiigehtchic, Northwest Territories (67° 260' N, 133° 450' W). The mean annual air temperature at these sites is -8.8 °C. The sites are evergreen needle forests dominated by *Picea mariana* (Mill.) BSP
and *Picea glauca* (Moench) Voss species. Site selection and physical characteristics of the sites are also described in Köster et al. (2017) and Aaltonen et al. (2019b).

Chronosequence sites were selected from the time since last burned with a stand replacing fire (in 1968, 1990, and 2012) and included a control site, where the last fire was more than 100 years ago. The date and boundaries of the fires were determined from geographic data from the Canadian Wildfire Information System (Natural Resources Canada). We visually corroborated
the geographic location of our sites with reported fire boundaries. Previous studies with these data (Köster et al., 2017; Aaltonen et al., 2019b, a; Zhou et al., 2019) classified the 1968 site as 1969, which we attribute from this site being classified by fire season, rather than the year of burn. For this manuscript we will refer to a site as a categorical variable by the year it was burned (2012, 1990, 1968) or the control site as "Control". Sites will be ordered by the fire year (2012, 1990, 1968, or Control).

At each site we measured soil temperature, fluxes of $CO_2$, microbial biomass assays, soil carbon, tree biomass (foliage,
branches, and stems), and other auxiliary measurements by establishing three different lines at each site, and within each line, three replicate plots (Köster et al., 2017). Additionally, at each plot soil samples were collected for further analysis in

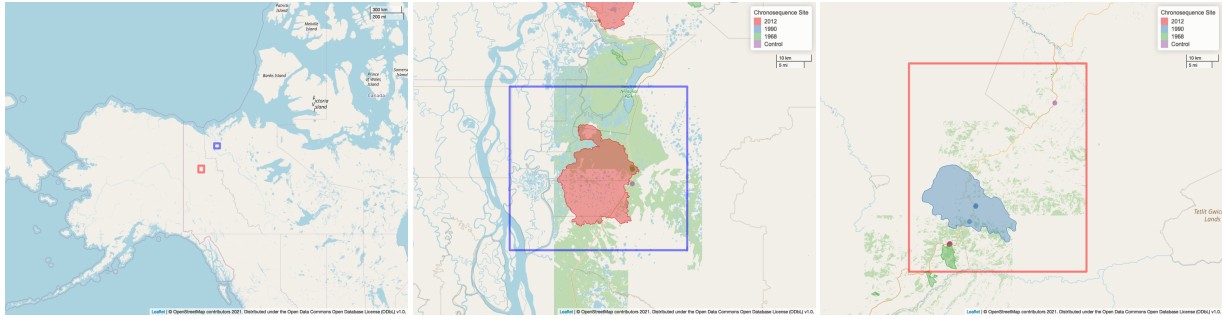

**Figure 1.** Map of chronosequence site locations in the Yukon and the Northwest Territories of Canada. In the two inset maps the boundaries of the fire areas are shown along with the location of the sampling sites (color coded the same as the fire areas). Maps provided by OpenStreetMap; © OpenStreetMap contributors 2021. Distributed under the Open Data Commons Open Database License (ODbL) v1.0.

incubation studies. Roots were excluded from incubation soils; we assume the measured respiration from these samples is $R_H$. The soil samples were incubated in 1, 7, 13 and 19 °C for 24 h and the respiration was measured from syringe samples taken at the end of each 24 h period. The method is described in more detail in Aaltonen et al. (2019b).

The field data measured total soil carbon in the top 30 cm, whereas the incubation data measured soil carbon to a given depth (5 or 10 cm). To minimize the effects between pool sizes between incubation and field data we applied a multistep process. First, for the soil carbon in each of the incubation samples we computed the cumulative proportion of soil carbon ($\mathrm{g\,C\,m^{-2}}$) to 30 cm. Second, at each plot we fit a saturating function to the cumulative proportion ($y = 1 - e^{-kD}$, where $y$ is the cumulative soil proportion at depth $D$). We assume that the amount of soil carbon deeper than 30 cm is negligible; by association we

assume a negligible amount of soil respiration beyond 30 cm. From the median ensemble average of fitted equations (Figure 2) we then computed the proportion of soil carbon up to a given depth (5 or 10 cm) at each site (Table 1). These proportions were then used for determining the amount of soil carbon at 5 or 10 cm for the field data.

    Incubation data also measured the available soil organic carbon extracted from incubation soils, denoted here as $C_A$, as described in Zhou et al. (2019). Briefly, soil dissolved organic C content was measured using total organic C analyzer (Shimadzu

TOC-V CPH, Shimadzu Corp., Kyoto, Japan) from soil extracts extracted with 0.5 M $K_2SO_4$. Microbial carbon used in the FireResp model was extracted using the chloroform fumigation extraction method (Beck et al., 1997). Briefly, three grams dry weight equivalent of soil was fumigated at 25 °C with ethanol-free chloroform for 24 h and extracted with 0.5-M $K_2SO_4$. The conversion factors, also known as the extraction efficiency, for estimating the microbial carbon is 0.45 (Beck et al., 1997). For the field data, we approximated $C_A$ as linearly associated with total soil carbon $C_S$ at a given depth, extrapolated from linear

regression in the incubation data (results not shown).

    For the field samples an estimate of root carbon $C_R$ was assumed to be proportional to total tree biomass collected at each plot (Härkönen et al., 2011; Neumann et al., 2020). A summary of all input variables is reported in Table 1.



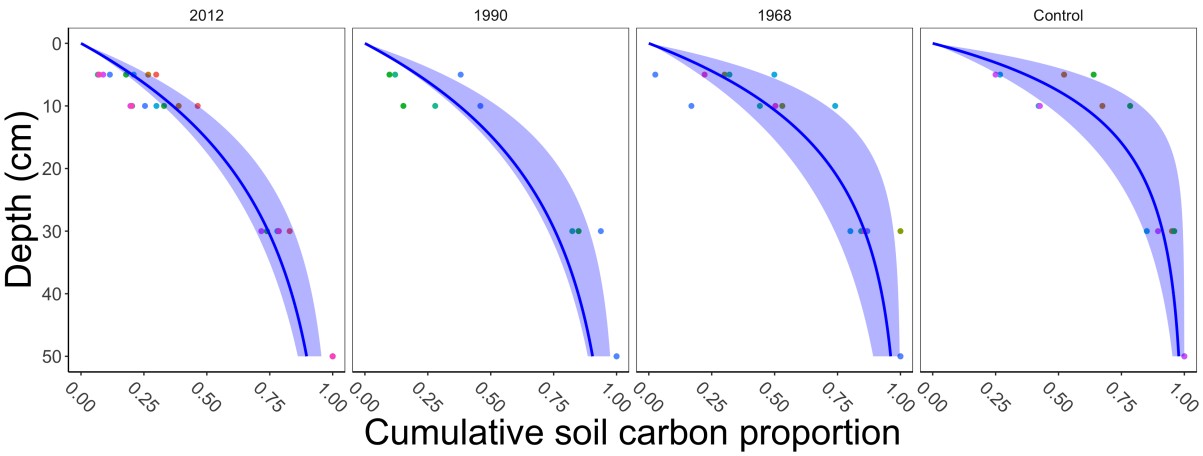

**Figure 2.** Summary plot of the cumulative proportion of soil carbon collected by depth. Each facet represents a different site in the chronosequence, and each of the dots represents a plot measurement. At each site we then fit a saturating function for each plot and then computed the ensemble average (median with 95% confidence interval, blue shading) from the fitted results.





| | Incubation Data | | | | | Field Data | | | | | | |
| --- | --- | --- | --- | --- | --- | --- | --- | --- | --- | --- | --- | --- |
| | Total $C_S$ proportion* | $C_S$† | $C_M$† | $C_A$† | $T_{soil}$‡ | $R_H$§ | $f_W$* | $C_S$† | $C_M$† | $C_A$† | $C_R$† | $R_S$§ |
| **2012** | | | | | | | | | | | | |
| 5 cm | 0.20 | 2669 ± 1474 | 10.5 ± 12.9 | 4.6 ± 5.7 | 1.13 ± 1.30 | 9.2 ± 3.0 | 0.36 ± 0.05 | 2592 ± 597 | 10.4 ± 13.1 | 4.8 ± 1.3 | 0 ± 0 | 0.91 ± 0.33 |
| 10 cm | 0.16 | 2348 ± 400 | 0.3 ± 0.2 | 2.1 ± 0.4 | 0.37 ± 0.30 | 6.4 ± 2.1 | 0.36 ± 0.06 | 2156 ± 392 | 0.3 ± 0.3 | 1.9 ± 0.3 | 0 ± 0 | 1.01 ± 0.24 |
| All depths | 0.36 | 2560 ± 1222 | 7.1 ± 11.5 | 3.7 ± 4.8 | 0.87 ± 1.13 | 8.1 ± 3.0 | 0.36 ± 0.05 | 2424 ± 555 | 6.5 ± 11.2 | 3.7 ± 1.8 | 0 ± 0 | 0.95 ± 0.29 |
| **1990** | | | | | | | | | | | | |
| 5 cm | 0.21 | 3684 ± 3327 | 9.5 ± 9.2 | 5.9 ± 6.4 | 1.80 ± 3.42 | 9.6 ± 0.4 | 0.41 ± 0.10 | 3666 ± 912 | 10.7 ± 7.8 | 5.8 ± 1.3 | 26 ± 21 | 2.03 ± 0.52 |
| 10 cm | 0.17 | 2302 ± 967 | 4.1 ± 4.7 | 3.4 ± 3.5 | 1.10 ± 1.48 | 7.5 ± 1.1 | 0.37 ± 0.09 | 2932 ± 989 | 9.0 ± 10.7 | 4.6 ± 1.9 | 16 ± 4 | 1.85 ± 0.50 |
| All depths | 0.38 | 3177 ± 2766 | 7.5 ± 8.2 | 5.0 ± 5.6 | 1.54 ± 2.85 | 8.8 ± 1.3 | 0.40 ± 0.10 | 3384 ± 974 | 10.0 ± 8.6 | 5.3 ± 1.6 | 22 ± 17 | 1.96 ± 0.50 |
| **1968** | | | | | | | | | | | | |
| 5 cm | 0.28 | 6392 ± 3520 | 48.4 ± 45.1 | 44.4 ± 34.6 | 1.73 ± 1.79 | 7.5 ± 2.4 | 0.46 ± 0.05 | 4565 ± 1372 | 27.8 ± 22.3 | 32.1 ± 9.3 | 1018 ± 312 | 2.97 ± 0.38 |
| 10 cm | 0.20 | 4047 ± 2139 | 5.4 ± 3.6 | 4.1 ± 3.2 | 0.31 ± 0.43 | 5.9 ± 0.9 | 0.43 ± 0.06 | 2620 ± 719 | 3.1 ± 1.1 | 1.9 ± 1.1 | 1015 ± 200 | 2.94 ± 0.43 |
| All depths | 0.48 | 5528 ± 3259 | 32.6 ± 41.4 | 29.6 ± 33.7 | 1.21 ± 1.59 | 6.9 ± 2.1 | 0.44 ± 0.05 | 3787 ± 1492 | 17.9 ± 21.0 | 20.0 ± 17.1 | 1017 ± 259 | 2.96 ± 0.38 |
| **Control** | | | | | | | | | | | | |
| 5 cm | 0.37 | 8210 ± 3842 | 74.5 ± 80.6 | 36.8 ± 25.4 | 5.70 ± 5.15 | 6.8 ± 0.8 | 0.53 ± 0.18 | 5789 ± 1893 | 54.4 ± 35.6 | 24.8 ± 10.0 | 1717 ± 199 | 2.16 ± 0.90 |
| 10 cm | 0.22 | 2377 ± 467 | 6.3 ± 4.0 | 4.9 ± 3.6 | 0.63 ± 0.44 | 4.3 ± 1.2 | 0.54 ± 0.16 | 3304 ± 1088 | 9.6 ± 4.9 | 3.3 ± 1.8 | 1604 ± 276 | 1.84 ± 0.99 |
| All depths | 0.59 | 5028 ± 3908 | 37.3 ± 63.6 | 19.4 ± 23.4 | 2.94 ± 4.27 | 5.3 ± 1.6 | 0.53 ± 0.16 | 4369 ± 1909 | 28.8 ± 32.1 | 12.6 ± 12.7 | 1653 ± 244 | 1.98 ± 0.93 |

**Table 1.** Summary of soil measurements for this study, organized by site in the chronosequence (2012, 1990, 1968, Control) and depth of measurement. Reported values are averages ± standard deviation of plots from three sample lines.

*: No units

†: Units are g C m$^{-2}$

‡: Units are °C

§: Units are g C m$^{-2}$ d$^{-1}$





## 2.2 Description of FireResp model

The FireResp model predicts plot-level soil respiration ($R_S$) and its components: autotrophic respiration ($R_A$), microbial
maintenance respiration ($R_M$), and microbial growth respiration ($R_G$). All respiration units are reported as g C m$^{-2}$ d$^{-1}$. The
FireResp model expresses respiration components with two primary functions; the different combinations of these functions
yields different submodels (described in detail below). First, we assumpe that $R_A$ and $R_M$ both follow an exponential $Q_{10}$
relationship (Eq. (1)):

$$R_X = k_X C_X f_W Q_{10,X}^{(T_{soil}-10)/10}. \tag{1}$$

Eq. (1) temperature is a commonly applied (empirical) paradigm for respiration, motivated by temperature dependencies of
enzymatic reactions (van't Hoff and Lehfeldt, 1898). This exponential temperature model is applied for $R_A$ and $R_M$, similar to
process models for these components at the ecosystem scale (Aber et al., 1997; Zobitz et al., 2008). Input variables for Eq. (1)
are soil temperature ($T_{soil}$; °C), volumetric soil moisture ($f_W$; %). The variable $C_X$ represents a soil carbon pool (g C m$^{-2}$).
For $R_A$ this $C_X$ equals root carbon ($C_R$); for $R_M$ this $C_X$ equals soil carbon ($C_S$) or microbe carbon ($C_M$) depending on
the type of submodel considered (e.g. Null, Microbe, Quality, Microbe-mult, or Quality-mult; all described below). Eq. (1)
has two parameters: $k_X$, the base rate of respiration (d$^{-1}$) for pool $C_X$, and $Q_{10,X}$ the temperature response of respiration
($Q_{10}$ value) (no units) for pool $X$. To aid the representation of model equations, we will write Eq. (1) as $R_X = g_X C_X$, where
$g_X = k_X Q_{10,X}^{(T_{soil}-10)/10}$. As an example, autotrophic respiration $R_A$ would be written as $R_A = g_R C_R$.

Second, we model microbial growth respiration ($R_G$) via Michaelis-Menten kinetics (Michaelis and Menten, 1913; Davidson
et al., 2006; German et al., 2012):

$$R_G = \epsilon \frac{\mu C_X C_M}{k_A + C_X}. \tag{2}$$

Eq. (2) arises from first-order microbial enzyme kinetics (Allison et al., 2010) under quasi-steady state assumptions (Keener
et al., 2009). In Eq. (2), $\epsilon$ is the efficiency converting substrate to microbe biomass (no units), $\mu$ is the maximum microbial
uptake rate (hr$^{-1}$), and $k_A$ (g C m$^{-2}$) represents the half-saturation rate, and $C_X$ represents the substrate for respiration.
Depending on the model variant $C_X$ may be total soil carbon ($C_S$) or available soil organic carbon ($C_A$), which represents
more labile carbon for ingestion by microbes.

The FireResp model has five different submodels which arise through different combinations of these functional representa-
tions of respiration. These submodels are slightly modified from a similar approach in Zobitz et al. (2008):

- **Null submodel:** Here soil maintenance respiration depends on soil carbon (so $R_M = g_S C_S$). Microbe carbon is not
considered in the Null submodel, so total soil respiration ($R_S$) is the sum of autotrophic and maintenance respiration
    (Eq. (3)):

$$R_S = R_A + R_M = g_R C_R + g_S C_S. \tag{3}$$





The Null submodel assumes a single soil carbon pool with respiration as only temperature dependent (Davidson et al., 1998; Reichstein and Beer, 2008), where there is only a single soil pool.

– **Microbe submodel:** Here maintenance respiration is proportional to microbial carbon, so $R_M = g_M C_M$. For growth respiration ($R_G$) total soil carbon ($C_S$) is the input for pool $C_X$ in Eq. (2). With these considerations total soil respiration is expressed in Eq. (4):

$$R_S = R_A + R_M + R_G = g_R C_R + g_M C_M + \epsilon \frac{\mu C_S C_M}{k_A + C_S} \tag{4}$$

The Microbe submodel is based on a two-pool soil-microbe model described in Sihi et al. (2016).

– **Microbe-mult submodel:** This submodel is structured similar to the Microbe model but with two modifications. First, growth respiration is not considered. Second, maintenance respiration is multiplied by a Michaelis-Menten factor:

$$R_S = R_A + R_M = g_R C_R + g_M C_M \cdot \frac{C_S}{k_A + C_S} \tag{5}$$

The Microbe-mult model is designed to be an intermediate model between the Null model and the Microbe model. The additional multiplicative factor is a heuristic designed to represent maintenance respiration as substrate limited by $C_S$.

– **Quality submodel:** This submodel is structured similar to the Microbe model, but for growth respiration ($R_G$) available soil organic carbon ($C_A$) is the input for pool $C_X$ in Eq. (2). Total soil respiration is expressed in Eq. (6):

$$R_S = R_A + R_M + R_G = g_R C_R + g_M C_M + \epsilon \frac{\mu C_A C_M}{k_A + C_A} \tag{6}$$

The Quality submodel is based on a multi-pool soil model that structures the soil into different pools based on the recalcitrance and turnover time of the soil parent material, similar to models by Bosatta and Ågren (1985). Inputs from
litterfall, enzymatic degradation, root turnover, or root exudation create a pool of available soil organic carbon ($C_A$) that can be incorporated into microbial biomass. While in this case $R_G$ is represented with Eq.(2), when constructing a dynamic model of soil would additionally include expressions for the transformation of each soil pool through enzymatic degradation and mineralization to a more recalcitrant pool (both under first-order kinetics).

    – **Quality-mult submodel:** This submodel is structured similar to the Quality model with two modifications (similar to
the modifications made in the Microbe-mult model). First, growth respiration is not considered. Second, maintenance respiration is multiplied by a Michaelis-Menten factor:

$$R_S = R_A + R_M = g_R C_R + g_M C_M \cdot \frac{C_A}{k_A + C_A} \tag{7}$$

Like the Microbe-mult model, the Quality-mult is a heuristic model designed to represent maintenance respiration as substrate limited by $C_A$.

Table 2 summarizes the different parameters for each model and their allowed ranges when estimating parameters.





| Name | Description (units) | Allowed Ranges |
|---|---|---|
| $Q_{10,M}$ | Microbe $Q_{10}$ (no units) | [1,5] |
| $Q_{10,R}$ | Root $Q_{10}$ (no units) | [1,5] |
| $k_R$ | Basal root respiration rate ($\mathrm{d}^{-1}$) | [0,1] |
| $k_M$ | Basal microbe respiration rate ($\mathrm{d}^{-1}$) | [0,0.1] |
| $k_A$ | Microbe half saturation rate ($\mathrm{g\,C\,m}^{-2}$) | [0, 100000] |
| $\mu$ | Microbial maximum uptake rate ($\mathrm{h}^{-1}$) | [0,100] |
| $\epsilon$ | Microbial efficiency (no units) | [0,1] |
| $k_S$ | Heterotrophic respiration rate ($\mathrm{d}^{-1}$) | [0,0.1] |
| $f$ | Scaling parameter for heterotrophic respiration* (no units) | [0.5,1.5] |
| $g_R$ | Basal root respiration rate*,† ($\mathrm{g\,C\,m}^{-2}\,\mathrm{d}^{-1}$) | [0,0.1] |

**Table 2.** Description of parameters used in this study for each model along with the allowed range.

∗: denotes a parameter for the Incubation Field Linear parameter estimation approach.

†: denotes a parameter for the Field Linear parameter estimation approach.

## 2.3 Parameter estimation routine

The different submodels (Null, Microbe, Quality, Microbe-mult, and Quality-mult) may be nonlinear with respect to the parameters. For parameter estimation we applied the Levenberg-Marquardt algorithm (Elzhov et al., 2016). The Levenberg-Marquardt algorithm optimizes an objective function, which in this case is the residual sum of squares between measured and modeled soil respiration $R_S$. The algorithm also requires (1) the Jacobian of the model to accelerate convergence to the optimum value, (2) an initial guess for parameters, (3) and bounds for all parameters.

The Levenberg-Marquardt algorithm may converge to a local (rather than global) optimum or the estimated parameter values may be at the boundaries of the allowed range. To ensure that parameter estimates converged to a global (rather than local optimum), initial parameter guesses for the method were drawn from a uniform distribution with reasonable bounds on parameters (Table 2). The Levenberg-Marquardt algorithm is implemented in R with the package nlsr (Nash, 2014; Nash and Murdoch, 2019).

We examined how submodel results are modified when including the incubation data. We first applied the incubation data to estimate parameters related to $R_H$, and then used the field data to estimate parameters related to $R_A$. This sequential approach reduces the number of parameters to be estimated with field data. To examine the effect of the incubation data on model results, we implemented a quasi-factorial design utilizing different combinations of field or incubation data:

(1) **Field**: All model parameters (e.g. $Q_{10,M}$, $k_M$, $k_A$, $\mu$, $\epsilon$, $k_S$, $Q_{10,R}$, and $k_R$; depending on the type of model) were estimated with the field data only.





| Parameter estimation approach name ↘ | Data for assimilation | |
| --- | --- | --- |
| | Incubation (for $R_H$) & Field (for $R_A$) | Field (for $R_A$ & $R_H$) |
| $R_A$ depends on $T_{soil}$ | Incubation Field | Field |
| $R_A$ independent of $T_{soil}$ | Incubation Field Linear | Field Linear |

**Table 3.** Relationship between the different parameter estimation approaches utilized for this study.

(2) **Field Linear**: Model parameters for $R_H$ (e.g. $Q_{10,M}$, $k_M$, $k_A$, $\mu$, $\epsilon$, $k_S$, depending on the type of model) are estimated with the field data. Rather than a $Q_{10}$ function for $R_A$ (Eq. (1)), for this approach $R_A$ equals $g_R \cdot C_R$, where $C_R$ is provided by the field data. We then estimated $g_R$ from the field data.

(3) **Incubation Field**: Two separate parameter estimations were applied. First model parameters for $R_H$ (e.g. $Q_{10,M}$, $k_M$, $k_A$, $\mu$, $\epsilon$, $k_S$ depending on the type of model) were first estimated with the incubation data. Next, autotrophic respiration parameters ($Q_{10,R}$ and $k_R$) were estimated from field data.

(4) **Incubation Field Linear**: Similar to the Incubation Field approach, parameters relating to $R_H$ were first estimated with incubation data. Next using these parameter estimates, heterotrophic respiration was computed from the corresponding field measurements (denoted here as $R_{H,field}$). Total soil respiration then equals $R_S = g_R \cdot C_R + f \cdot R_{H,field}$, with $R_A = g_R \cdot C_R$ and $R_H = f \cdot R_{H,field}$. We then estimated $f$ and $g_R$ from the field data.

Table 3 shows the relationship between the different parameter estimation approaches studied.

Table 4 lists the parameters estimated for each model and parameter estimation approach. Data used for parameter estimation consisted of combinations from five different categories of sites (2012, 1990, 1968, Control, or all sites together) and 3 different depths (5 cm, 10 cm, or both depths together). Additionally with the four different parameter estimation approaches (Field, Field Linear, Incubation Field, and Incubation Field Linear) and five different models (Null, Microbe, Microbe-mult, Quality, and Quality-mult), 300 separate parameter estimations were computed.

When parameters were estimated either with the incubation data, Field approach, and Field Linear approach, we applied 1000 iterations of the Levenberg-Marquardt algorithm. Following those iterations, we then applied two filters to reduced post-processing computational time. First, we filtered parameter sets where the computed residual sum of squares was contained within the 25[th] and 75[th] percentiles, and next we removed instances where the set of parameters were duplicated. For the Incubation Field and Incubation Field Linear approaches then used the filtered parameter set from the incubation data for subsequent estimation of the remaining parameters with field data.

## 2.4 Model evaluation

We applied two different approaches to evaluate the reasonableness of a model-data fit. The first approach relied on Taylor diagrams (Taylor, 2001), which facilitates intercomparison between models when compared to measured values (in this case $R_S$). The Taylor diagram is structured as a polar coordinate plot; here the radius $\nu$ is the normalized ratio between modeled





| Parameter estimation approach → | Field | Field Linear | Incubation Field | Incubation Field Linear |
|---|---|---|---|---|
| **Null submodel ($R_S = R_A + R_M$):** | | | | |
| $R_A$: | $Q_{10,R}, k_R$ | $g_R$ | $Q_{10,R}, k_R$ | $g_R$ |
| $R_M$: | $Q_{10,M}, k_M$ | $Q_{10,M}, k_M$ | $\boldsymbol{Q_{10,M}, k_M}$ | $\boldsymbol{Q_{10,M}, k_M}$ |
| | | | | $f$ |
| Number of parameters: | 4 | 3 | 4 | 4 |
| | | | | |
| **Microbe & Quality submodels ($R_S = R_A + R_M + R_G$):** | | | | |
| $R_A$: | $Q_{10,R}, k_R$ | $g_R$ | $Q_{10,R}, k_R$ | $g_R$ |
| $R_M$: | $Q_{10,M}, k_M$ | $Q_{10,M}, k_M$ | $\boldsymbol{Q_{10,M}, k_M}$ | $\boldsymbol{Q_{10,M}, k_M}$ |
| $R_G$: | $k_A, \mu, \epsilon$ | $k_A, \mu, \epsilon$ | $\boldsymbol{k_A, \mu, \epsilon}$ | $\boldsymbol{k_A, \mu, \epsilon}$ |
| | | | | $f$ |
| Number of parameters: | 7 | 6 | 7 | 7 |
| | | | | |
| **Microbe-mult & Quality-mult submodels ($R_S = R_A + R_M$):** | | | | |
| $R_A$: | $Q_{10,R}, k_R$ | $g_R$ | $Q_{10,R}, k_R$ | $g_R$ |
| $R_M$: | $Q_{10,M}, k_M, k_A$ $R_G$: | $Q_{10,M}, k_M, k_A$ | $\boldsymbol{Q_{10,M}, k_M, k_A}$ | $\boldsymbol{Q_{10,M}, k_M, k_A}$ |
| | | | | $f$ |
| Number of parameters: | 5 | 4 | 5 | 5 |

**Table 4.** Listing of parameters estimated with each submodel and parameter estimation approach. Parameters in bold-face font (Incubation and Incubation Field Linear approaches) were estimated from the incubation data first, followed by all remaining parameters with the field data.

and measured standard deviation $\sigma_{model}/\sigma_{measured}$ and the angle $\theta$ corresponding to the correlation coefficient $r$ for measured

and modeled $R_S$. Two comparisons can be visually inferred from the Taylor diagram. First, the point located at $(\nu, \theta) = (1, 0)$ represents a set of modeled values of $R_S$ that perfectly match measured $R_S$. Values of $\nu$ less than unity indicate that modeled $R_S$ has less variability. Second, the distance from a point on the diagram to $(\nu, \theta) = (1, 0)$ is the centered pattern root mean square distance. Concentric circles from the point $(\nu, \theta) = (1, 0)$ help assess the compare the centered pattern root mean square distance for modeled results.

A second approach relies on Akaike's Information Criteria (AIC) (Akaike, 1974). The AIC is defined as $-2 \cdot LL + 2 \cdot p$, where $LL$ is the log-likelihood and $p$ the number of parameters in the model. The submodel with the lowest AIC is defined as the best approximating model for the data. We apply the AIC to compare across submodels for a parameter estimation approach to control for sample size effects in the AIC.

# 3 Results

With the different combinations of measurements (incubation or field measurements), models (Section 2.2) and parameter estimation approaches (Section 2.3) we have over 300 different estimates of the parameters. Parameter estimates were evaluated



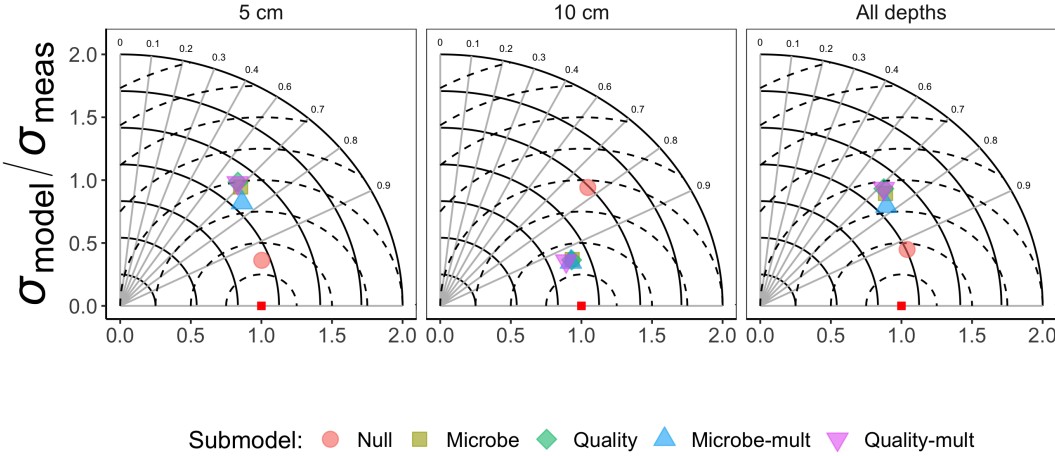

**Figure 3.** Taylor diagram for optimization for incubation data. Columns in the facetted plot represent the depth of the data used for parameterization (5 cm, 10 cm, or All depths), rows refer to the chronosequence site. Radii represent the normalized standard deviation between a submodel value of $R_S$ to measured $R_S$, angles the correlation coefficient $r$ (labeled). The dashed concentric circles represent contours (increments 0.25) for the normalized centered pattern root mean square distance.

based on the summary distributions of modeled $R_A$, $R_H$, and $R_S$. Results were evaluated for both their reasonableness to produce estimates of $R_A$ and $R_H$ as well as the comparisons between measured and modeled $R_S$ for incubation and field data (Taylor diagrams).

Figure 3 shows the Taylor diagram for incubation data, faceted by the depth of soil data used for parameter estimation (5 cm, 10 cm, or both). We combined data from all sites in the chronosequence to make these comparisons. In general most models had high correlation coefficients ($\approx 0.7 - 0.9$); combining all the sites together did not improve the model-data comparisons. Figure 4 is structured similar to Figure 3, but instead uses field data with the different approaches to optimization.

We used sparkline tables to summarize and compare the panoply of parameter statistics (Figure 5) and model statistics

(adjusted $R^2$ and AIC, Figure 6). In particular column (parameter) in Figure 5, the vertical axis is scaled to the ranges of the parameters in Table 2; the horizontal axis is ordered by the time since disturbance (2012, 1990, 1968, or Control sites). For ease of presentation, Figure 5 displays results from the Incubation Field Linear approach at 5 cm; all the model results are presented in the Supplementary Information. Figure 5 also denotes edge-hitting parameters (defined here as within a tenth of percent of the range of the parameters) as separate colors. In contrast, Figure 6 structures each sparkline plot by the submodel studied

(Null, Microbe, Quality, Microbe-mult, and Quality-mult), facilitating comparisons between models for a given parameter estimation and depth of data used in the parameter estimation. In Figure 6, sparkline plots for adjusted $R^2$ or AIC values are all scaled respectively the same for each statistic. The models with the largest adjusted $R^2$ or lowest AIC value are denoted as separate colors.



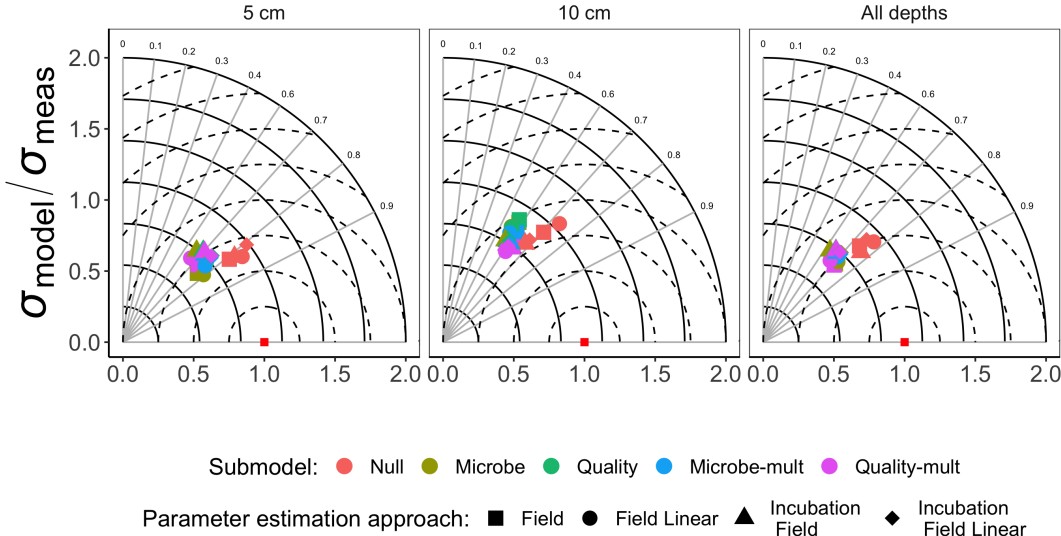

**Figure 4.** Taylor diagram for optimization for field data. Columns in the facetted plot represent the depth of the data used for parameterization (5 cm, 10 cm, or combined), rows refer to the chronosequence site and the year the site was burned.

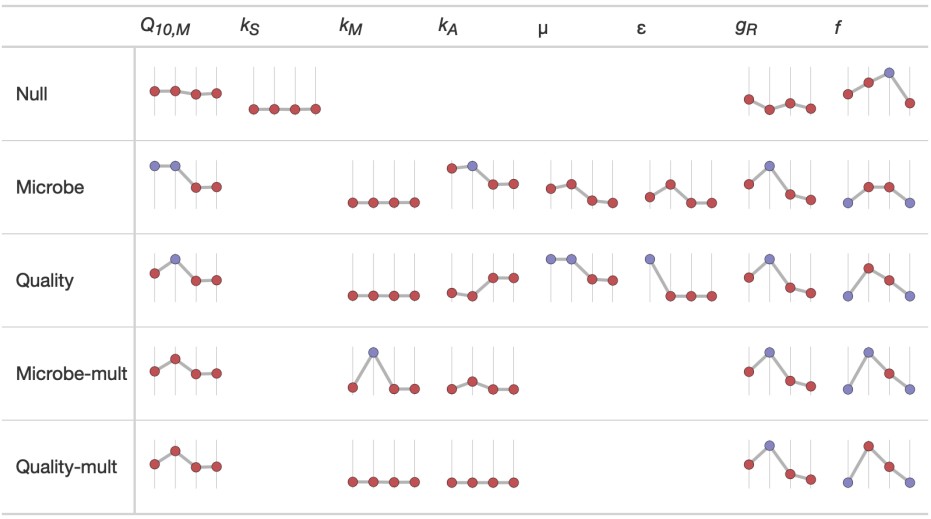

**Figure 5.** Median values of parameter estimates for different models using the Incubation Field Linear approach at 5 cm depth. The horizontal axis on each sparkline plot is arranged by the year since the burn sites in the chronosequence (2012, 1990, 1968, or Control). In each column the vertical axis scale is the same. Edge hitting parameters are denoted with the blue coloring.





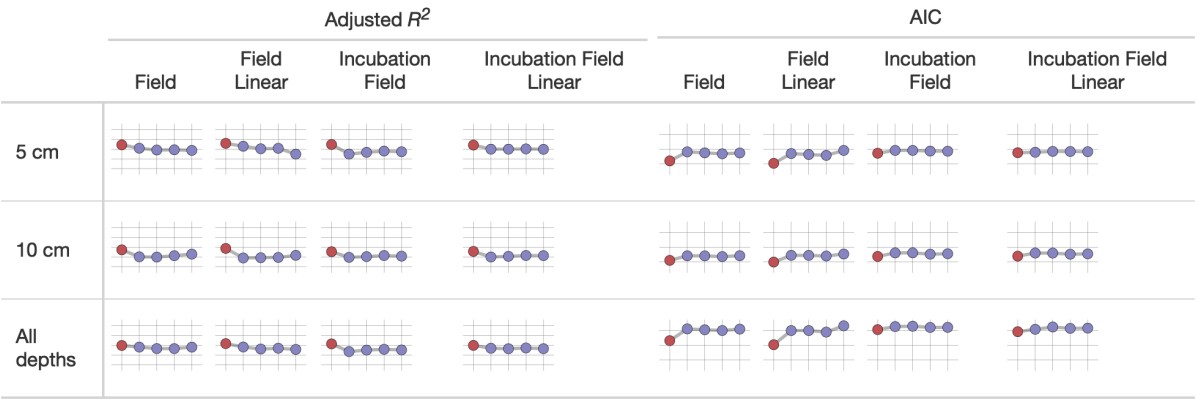

**Figure 6.** Median values of the adjusted $R^2$ and AIC from different parameter estimation approaches (Field, Field Linear, Incubation Field, and Incubation Field Linear) using measurements made at given depth. The horizontal axis on each sparkline plot is arranged by the different models studied (Null, Microbe, Quality, Microbe-mult, and Quality-mult). For the adjusted $R^2$ sparkline plot, the vertical axis ranges between 0 to 1, with gridlines every 0.25 units. The submodel with the highest adjusted $R^2$ value is denoted with red coloring. For the AIC plots, the vertical axis ranges from 50 to 150, with gridlines every 50 units. The submodel with the lowest AIC is denoted with red coloring.

We computed $R_A$, $R_H$, and the proportion of soil respiration due to autotrophic respiration ($p_A = R_A/(R_A + R_H)$) for each
parameter set generated through the parameter estimation routine (Section 2.3). We then computed summary statistics from the distribution of $R_A$, $R_H$, $p_A$ for each parameter estimation approach. Summary results for the median of these distributions for $R_A$ and $R_H$ are shown in Figure 7, organized by the parameter estimation approach. Additionally the red shading in Figure 7 shows the minimum and maximum ranges of measured $R_S$ (lines), first or third quarters (boxes), and median $R_S$ for comparison. Figure 7 visually displays no significant difference in patterns of $R_A$ and $R_H$ by the depth of the soil data used
for parameter estimation (5 cm, 10 cm, or both depths together).

Figure 8 is structured similar to Figure 7, but shows $p_A = R_A/(R_A + R_H)$, which facilitates better comparison across the different types of approaches to estimate parameters. For comparison, the green boxes show the predicted values of $p_A$ based on $R_A$ and $R_H$ data reported in Figure 1 of Ribeiro-Kumara et al. (2020b) (available through Mendeley; Ribeiro-Kumara et al. (2020a)). We computed the predicted values of $p_A$ from a loess fit using years since disturbance and $p_A$ as variables.

**4   Discussion**

Soil models that directly incorporated microbe carbon produced patterns of $R_A$ and $R_H$ that increased from the time since the fire (Figure 7). As these patterns also conform to changes in root carbon (which was proportional to tree biomass, Table 1), we have initial support for our two primary hypotheses: (1) autotrophic respiration should be positively associated with the time since disturbance because of changes in aboveground foliar vegetation from forest succession and (2) when corroborated with
observational data, soil models that include soil microbial carbon will better replicate expected patterns for soil respiration

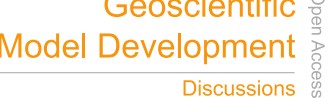

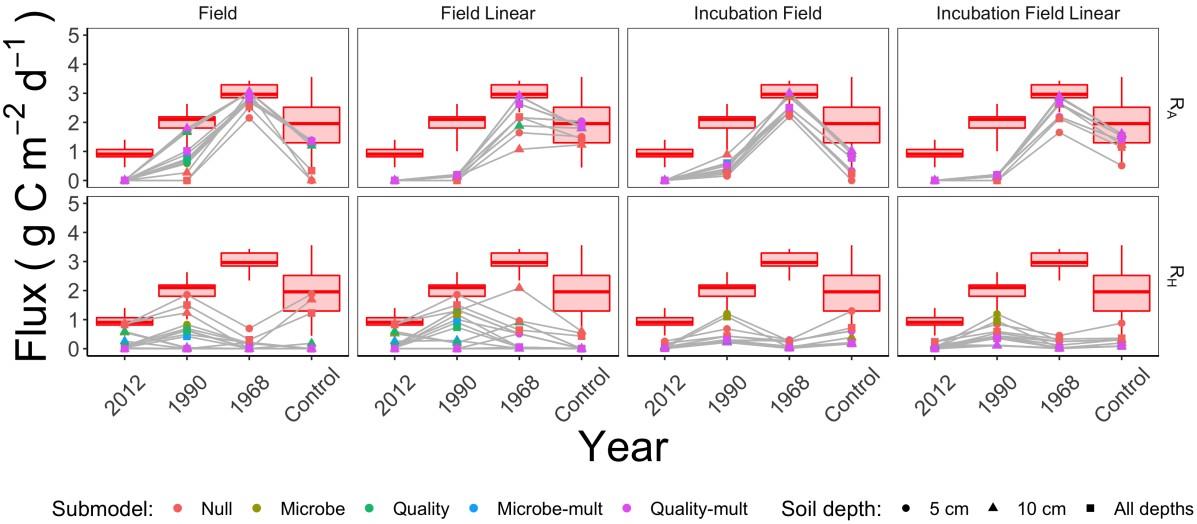

**Figure 7.** Median modeled fluxes of $R_A$ and $R_H$ from different parameter estimation approaches (Field, Field Linear, Incubation Field, Incubation Field Linear), soil depth data used for parameter optimization (5 cm, 10 cm, or both depths together) and submodel (Null, Microbe, Quality, Microbe-mult, and Quality-mult). The boxplot shows measured ranges of $R_S$ at each site in the chronosequence.

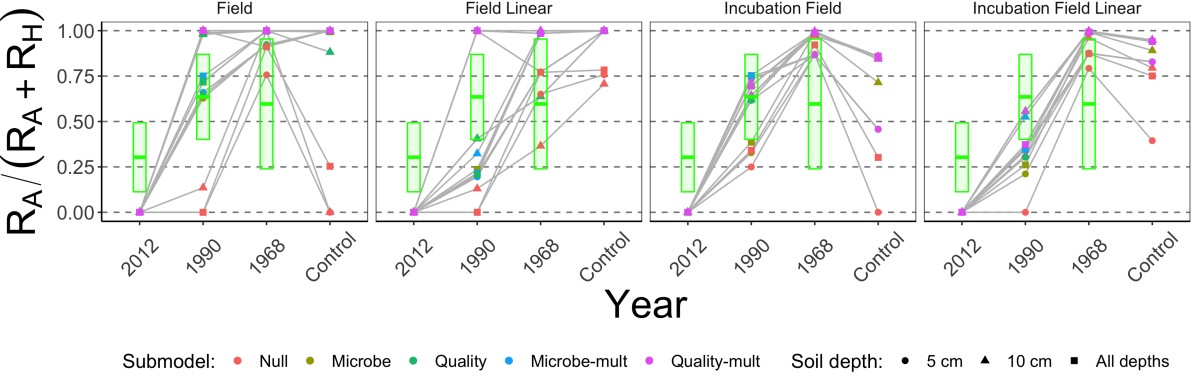

**Figure 8.** Median contribution of the proportion of autotrophic respiration ($p_A = R_A/R_S$) from different parameter estimation approaches (Field, Field Linear, Incubation Field, and Incubation Field Linear), soil depth data used for parameter optimization (5 cm, 10 cm, or both depths together) and model (Null, Microbe, Quality, Microbe-mult, and Quality-mult). The crossbar plot shows predicted values of $p_A$ with twice the standard error from data reported in Figure 1 in Ribeiro-Kumara et al. (2020b).




components across the chronosequence. We will further evaluate the two hypotheses through subsequent analysis of the data used for parameter estimation, parameter estimation approaches, and the soil respiration models.

### 4.1 Evaluation of datasets for parameter estimation

We had two categories of datasets for this study: the type of data (incubation or field data) or the depth at which measurements
were made (5 cm, 10 cm, or both depths together). This controlled experimental design is also represented in the Taylor diagrams (Figure 3) which shows, comparatively, a centered pattern root mean square distance (distance between a point on the Taylor diagram and $(\nu, \theta) = (1, 0)$) ranging from 0.25 - 1 and $r$ ranging 0.7 - 0.9. For the field data (Figure 4), the centered pattern root mean square distance ranged from 0.5 - 1 and $r$ 0.4 - 0.8. We attributed the difference between Figures 3 and Figure 4 is that the range of soil temperatures from the incubation experiments spanned 1 - 19 °C, allowing for a wider range
to characterize any exponential temperature profile. In contrast, field measurements ranged from 4 - 9 °C (Table 1). For both Figure 3 and Figure 4 the 5 cm depth had higher values for $r$ and a smaller centered pattern root mean square distance compared to the 10 cm depth.

We did not find any noticeable site differences in submodel outputs depending on the depth of the soil used for data assimilation (5 cm, 10 cm, or both depths together; Figures 3, 4, Figure 6). While soil model parameters (such as $Q_{10}$) are expected
to vary with soil depth (Pavelka et al., 2007; Graf et al., 2008; Pumpanen et al., 2008) we did not observe any significant depth-dependent differences in parameter estimates (see the figures in the Supplementary Information). The primary reason for this result is that the inter-site variability is larger than the variability by depth at a given site (Table 1 and Figure 2). We also did not find any improvements in our results when all data from sites were pooled together (Figure 7 and Figure 8). From these conclusions we will limit the discussion to evaluating model results generated from data at the 5 cm depth.

### 4.2 Evaluation of parameter estimation approaches

We cannot eliminate a parameter estimation approach (Field, Field Linear, Incubation Field, or Incubation Field Linear) simply by the magnitude of the estimated fluxes $R_A$ (Figure 7). Measured autotrophic respiration in actively growing high-latitude boreal forests (Bond-Lamberty et al., 2004; Vogel et al., 2005, 2014; Pumpanen et al., 2015) or inferred from synthesis studies (Ribeiro-Kumara et al., 2020b; Morgan et al., 2021) can range from 0.5 - 4 g C m$^{-2}$ d$^{-1}$. The modeled values of $R_A$ for all
the parameter estimation approaches are within that range.

While there is no universal pattern to $R_H$ following forest fire disturbances (Ribeiro-Kumara et al., 2020b), we have reason to believe the near-zero modeled values for $R_H$ for the 1968 site in Figure 7 may be an underestimate. For our sites we expect modest, and perhaps decreasing (but not zero), changes in $R_H$ from the time of disturbance due to three reasons. First, factors influencing recovery of $R_H$ are burn severity or intensity (Meigs et al., 2009; Hu et al., 2017) and decomposition of
pyrogenic litter (Kulmala et al., 2014; Muñoz-Rojas et al., 2016). The fires at our sites combusted a significant amount of soil organic matter (Köster et al., 2017) resistant to decomposition (Knicker, 2007; Aaltonen et al., 2019a), thereby minimizing any increases in $R_H$ from the decomposition of labile litter. Additionally, from this chronosequence, Aaltonen et al. (2019b) reported increased temperature sensitivity ($Q_{10,M}$) in recently burned sites, but this was tempered by decreases in soil organic





matter quality (Aaltonen et al., 2019a). Second, as succession occurs, the increase in aboveground vegetation insulates the

soil, decreasing the active layer and thereby decreasing $R_H$ (Köster et al., 2017). Third, at the same chronosequence sites Zhou et al. (2019) found constant C:N:P and fungal to bacterial ratios for microbes, indicating homeostatic regulation of the microbial community. The cumulative effect of these confounding factors may translate into $R_H$ remaining constant across the chronosequence.

Our models implicitly assumed an increasing exponential relationship between temperature and respiration. The tempera-

ture sensitivity of respiration ($Q_{10}$) across ecosystems can vary (usually around 2-5) (Chen and Tian, 2005; Wang et al., 2006; Bond-Lamberty and Thomson, 2010) and is generally expected to be greater than 1, but the $Q_{10}$ value may decrease as soils warm (Niu et al., 2021). Some degree of additional variability is expected when considering the biochemical or thermodynamic foundations of respiration (Lloyd and Taylor, 1994; Ito et al., 2015), the methodological approach used to measure soil respiration (Ribeiro-Kumara et al., 2020b), or variation in the soil organic matter supply (Davidson et al., 2006).

However, an increasing exponential relationship between temperature and respiration may not be robustly supported with observed data at the chronosequence sites. The forest fires at each site burnt a large portion of soil organic matter and killed the roots. Immediately following a fire, $R_S$ will be lower even if there are higher soil temperatures. In late-successional forests, the soil is colder and the active layer depth is smaller, even though there may be more soil respiration due to higher amounts of roots and soil organic matter; we observed such patterns across the chronosequence. The 2012 and 1990 sites had the highest values

of $T_{soil}$ (Table 1) but the lowest overall respiration (Figure 7). Across the chronosequence, scatterplots of respiration with temperature had a null or a negative relationship with temperature (results not shown). Empirically the negative association of respiration with temperature would imply a $Q_{10}$ value less than unity. As a result, to compensate for these opposing tendencies the $R_H$ parameters tend to be edge-hitting (Figure 5 and Supplementary Information).

We recommend either the Incubation Field or Incubation Field Linear parameter estimation approach for two reasons. First,

values of the proportion of the respiration that is autotrophic ($p_A = R_A/(R_A + R_H)$, Figure 8) for the Field or Field Linear approaches are unexpectedly and unrealistically large, attributed to the variation in $R_H$ (Figure 7). As a baseline, Hanson et al. (2000) reported values of $R_A/(R_A + R_H)$ to be approximately 0.50, which has also been supported in meta-analyses (Soil Respiration Database, Bond-Lamberty and Thomson (2010)). Second, the Incubation Field and Incubation Field Linear approaches in Figure 8 show a temporal pattern in $p_A$ similar to patterns reported in Bond-Lamberty et al. (2004) and the

predicted $p_A$ inferred from Ribeiro-Kumara et al. (2020b). The modeled values of $p_A$ are larger in late successional sites (.75 - 1), which may be an effect of the timing of field collection (August) when $R_A$ is at a seasonal peak (Bond-Lamberty et al., 2004; Pumpanen et al., 2015).

## 4.3 Evaluation of hypotheses

Our first hypothesis concerned the dependence of $R_A$ on tree biomass. We developed this hypothesis from our previous studies,

which concluded tree biomass was a key factor explaining patterns of soil respiration across the chronosequence (Köster et al., 2017; Aaltonen et al., 2019a, b). For all models and the Field Linear or Incubation Field Linear parameter estimation approaches, $R_A$ is proportional to $C_R$, which is proportional to tree biomass. Values of $C_R$ increase across the chronosequence



(Table 1). However even with this proportional association, the results in Figure 7 indicate less support for our first hypothesis for two reasons. First, some modeled values $R_A$ at the 1990 site are higher than expected, especially given the association with

$R_A$ to $C_R$. Since $C_R$ is still comparatively low at this site, we might expect $R_A$ (and by association $p_A$) to be near zero as well. Additionally, the near-zero values of $R_A$ are not a consequence of poorly-defined parameter estimates. The autotrophic respiration parameters are not overwhelmingly edge hitting (Figure 5 or Supplementary Information), indicating appropriately defined parameter bounds in Table 2. Second, and perhaps more importantly, all parameter estimation approaches in Figure 7 predict $R_A$ to decrease between the 1968 and Control sites. The modeled decreases in $R_A$ are a result of observed decreases

in $R_S$ (Figure 7) as $C_S$ increases. To compensate for this estimated parameters $k_R$ or $g_R$ decrease across the chronosequence sites (Figure 5 or Supplementary Information). The patterns to $k_R$ or $g_R$ may be due to the parameter estimation routine compensating the confounding effects of increasing $C_R$ with decreasing $R_S$. In summary, even though there is evidence for association between $R_A$ and tree biomass in earlier chronosequence sites (2012 and 1990 sites), additional work is needed to explain reasons for the decline in $R_A$ for later chronosequence sites (1968 and Control sites). Future work could quantify field

estimates of root mass, production, and turnover (Kalyn and Van Rees, 2006; Steele et al., 1997) to corroborate the values of $C_R$ used here and with the estimated decreases in $k_R$ across the chronosequence.

Our second hypothesis concerns the structural representation of soil respiration for soil models. Our submodels are arranged on a continuum of complexity (Null, Microbe, Quality, Microbe-mult, or Quality-mult). When parameterizing more complex models parameters may be non-informative and/or edge-hitting (Zobitz et al., 2011). Reducing parameter dimensionality is a

key consideration for model-data assimilation in the carbon cycle (Tang and Zhuang, 2008; Luo et al., 2009; Kraemer et al., 2020). Considering the Incubation Field Linear approach only, across the range of models the Microbe submodel had the smallest percentage of edge-hitting parameters (10%), ranging from 30 - 50% for the other models.

While the AIC suggests a preference towards the Null submodel, we do not believe it is a sufficient criterion to choose it over the Microbe and Quality submodels. There was no noticeable improvement with the Null submodel in the Taylor diagrams for

the field data (both in the values of $r$ and the centered pattern root mean square difference. Figure 4) or with the adjusted $R^2$ or AIC values (Figure 6). While all models could not account for a majority of the variance in observed soil respiration (the adjusted $R^2$ values in Figure 6 ranged from from .25 - .61), no submodel significantly improved the adjusted $R^2$ or AIC. In other words, the model statistics indicated the parameter estimation approaches all performed similarly.

A design constraint was to construct models with the greatest potential to be fully parameterized from the collected data.

For the Quality-mult and Microbe-mult models, $k_A$ was estimated at the lower end of its range (Figure 5), essentially reducing these models to the Quality and Microbe models respectively. Even though we cannot definitively conclude which of the two models (Quality or Microbe) is the better approximating model, we recommend that some consideration of microbial growth and maintenance respiration be considered using Michaelis-Menten kinetics as a starting point (Davidson et al., 2006). Several frameworks already exist for incorporating Michaelis-Menten kinetics (Todd-Brown et al., 2012) or substrate quality

degradation (Bosatta and Ågren, 1991, 2002). Continuous (daily or sub-daily) soil respiration measurements could better support more complex soil models (Rayment and Jarvis, 2000; Subke et al., 2006; Subke and Bahn, 2010; Phillips et al., 2011;





Pumpanen et al., 2015; Zhang et al., 2015). Each of the models could be incorporated into a dynamic model of ecosystem carbon cycling (Zobitz et al., 2008) that also include temporal changes of permafrost active layer depth (Zhu et al., 2019).

## 5 Conclusions

We examined the ability to parameterize a range of soil respiration models using data collected from a fire chronosequence. Importantly we found support for parameterizing a more complex submodel to replicate patterns in soil respiration and its components across a fire chronosequence. Separate analysis of soils with incubation experiments reduces the number of parameters to be estimated, however care must be taken in scaling incubation studies to field measurements.

For these high-latitude sites, future work could couple the models here to more continuous measurements of soil temperature
along with a dynamic active layer depth model (Zhu et al., 2019). These modeling approaches could examine the effects of gross primary productivity on soil respiration components (Zhuang et al., 2002; Pumpanen et al., 2003; Vargas et al., 2010; Pumpanen et al., 2015; Phillips et al., 2017). For sites that cannot be instrumented continuously (such as the ones studied here), this model data integration could be supported with periodic surveys of aboveground biomass and other remote sensing data (Neumann et al., 2020).

*Code and data availability.* Code and data necessary to reproduce all results are available through GitHub https://github.com/jmzobitz/FireResp and archived in Zenodo (Zobitz et al., 2021).

Supplementary Information includes sparkline parameter estimates (similar to Figure 5) for all approaches and depths examined.

*Author contributions.* Co-authors Zobitz and Pumpanen conceived the ideas for the research project, Co-authors Pumpanen, Köster, Beninger
collected the field data. Co-authors Aaltonen and Zhou analyzed the incubation data. All authors contributed to evaluating the results and the writing of the manuscript.}

*Competing interests.* The authors declare no competing interests.

*Acknowledgements.* Co-author Zobitz was funded by the Fulbright Finland Foundation and Saastamoinen Foundation Grant in Health and Environmental Sciences. This work was funded by the Academy of Finland (project numbers 286685, 294600, 307222, 327198, and 337550).
Travel funding was provided from EU InterAct (H2020 Grant Agreement No. 730938). Co-author Zobitz acknowledges B. S. Chelton for helpful discussion on this manuscript.



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
