# Peer review of "Evaluating an exponential respiration model to alternative models for soil respiration components in a Canadian wildfire chronosequence (FireResp, v1.0)"

_Geoscientific Model Development, 2021_

## Author Response (AR1)

We sincerely thank both reviewers for their careful and insightful comments and their appreciation of our manuscript. We have prepared a revised manuscript that incorporates their suggestions for improvement.

In addition to these edits and modifications described below we have also re-read the manuscript and as needed (1) made small editorial, grammar, spelling, and phrasing corrections, (2) increased the size of the figure annotations, and (3) revised figure and table captions to be more complete (as noted by Reviewer 1).

Below we address each of their comments (numbered individually) with our response *in italics*, where the heading X.Y refers to comment Y from reviewer X. In our responses we use the line numbers from the revised manuscript.
* * *
Reviewer 1

This manuscript describes a model-fitting and evaluation exercise that is unusual in a number of respects—in particular, the concurrent use of lab and field data, and the variety of models along a spectrum of complexity that were evaluated. This is an interesting and important topic, given our uncertainty about how disturbance affects soil carbon cycling and how best to model it. The ms is generally well written and interesting, and technically strong.

There are some problems. I'm confused by some points of the methods (see specific comments below); the text could do a better job of citing previous published work on a number of key points; and some of the figures and tables need better (more complete) captions.

Still, overall this is a very strong and interesting manuscript—nice job!—and these are only moderate changes that should help improve the clarity of the text.

*We have revised the figure and table captions to be more complete.*

Specific comments

1.1)  Lines 21-22: might mention fungi as an addition source

   *In line 23 We have mentioned fungi as an additional source with a reference to Anderson and Domsch (1973)*

1.2)  39: and to benchmark the models themselves, of course

   *In line 40 We have included "Observations of overall soil respiration can be linked with process-based soil models to estimate (and perhaps benchmark) $R_A$ and $R_H$."*

1.3)  45-52: Famiglietti et al. (2020)
   https://bg.copernicus.org/preprints/bg-2020-478/bg-2020-478.pdf might be a

good citation, re evaluating multiple models of different complexity. Also perhaps Shiklomanov et al. (2020) http://dx.doi.org/10.1111/gcb.15164

*Thank you for these references, we have incorporated both of these into this revision in lines 50 and 54.*

1.4)    62 and 254: "corroborated" isn't the word you want, I don't think. "tested against"?

*Yes - we agree - and have changed this in lines 65 and 265.*

1.5)    98: "incubation data" didn't measure; reword

*In lines 94 and 107 we have reworded this to "The incubation data included measurements of"*

1.6)    Table 1: define all parameters (Cs, etc)

*Based on similar comments from both reviewers we have revised this table to define all the parameters in the legend, removing the need for superscripts on the column headers. We also noticed that we had erroneously swapped the column labels for $T_{soil}$ and $R_H$ (but not the values). We apologize for this error.*

1.7)    112: "assume"

*Thank you for this correction.*

1.8)    114: Eq. 1 says that $f$w is volumetric soil water (%, so 0-100). First, this implies that respiration increases endlessly with increases in water, which can't be true, as soil anoxia will start to limit Rs at higher values. Second, I'm confused why $f$w doesn't appear in any subsequent equation. I see "$f$" in Table 2 but it's described simply as a scaling parameter.

*For this revision we have implemented the empirical function of soil moisture limitation described in Moyano et al. (2013), where r(fW) = 3.11 fW - 2.42 fW^2, which shows anoxia at higher soil moisture proportions. We compare this new*

*implementation (red curve below) to the linear scaling below (blue line below):*

[Figure]

*The green shading represents the range of soil moisture proportions found in our sites (Table 1). With the Moyano implementation this predicted higher values of $R_S$ compared to the previous version (Figure 7), but the pattern in the proportion of $R_A$ (Figure 8) is similar to the previous version. We've revised the text accordingly with this new implementation in Lines 118 - 128.*

*The parameter f in Table 2 was a factor that we introduced to scale between the incubation and field measurements. By including the function $r(f_W)$ in this revision we believe there is less chance of confusion between f and $f_W$.*

1.9) 200: "to reduce"

*At line 209 we revised this to: "Following these iterations we reduced post-processing computational time in two ways."*

1.10) 201: "we filtered" means you included, not excluded? Somewhat confusing given that the second part of the sentence describes excluding data

*In lines 209 - 213 we have clarified this phrasing: "Following these iterations we reduced post-processing computational time in two ways. First, duplicated parameter sets were reduced to a single instance. Second, we excluded parameter sets where the residual sum of squares was outside the 50% centered confidence interval. For the Incubation Field and Incubation Field Linear approaches, we used these filtered parameter sets for subsequent estimation of the remaining parameters with field data."*

1.11) Figure 5 caption: define "edge hitting"

*We have revised the phrase in this caption to "Edge-hitting parameters (defined here as within a tenth of percent of the allowed parameter range) are denoted with the blue coloring."*

1.12)   Figure 7: what do the grey lines show?

*We have added to this caption: "The grey lines are used as a guide to show the chronosequence trend for a particular parameter estimation approach and soil depth."*

1.13)   331: "patterns of…compensating for"

*Thank you for this correction in Line 343*

1.14)   348: Sulman et al. (2018) http://dx.doi.org/10.1007/s10533-018-0509-z might be an appropriate and useful citation here

*We have added in Lines 360 - 363 "This model result similarity conforms to a study by Sulman et al. (2018), which synthesized a range of experimental data with different types of process-based models to predict long-term soil organic carbon storage."*
* * *
Reviewer 2

This manuscript is an interesting paper presenting a qualitative evaluation of soil respiration models and their parameter estimation process. However some sentences or statements need additional clarification. Some extra work is also needed on figures in order to facilitate their reading and understanding.

Moreover, there is no mention and introduction of the FireResp model in the abstract or the introduction while it is mentioned in the title.

*We added to the abstract starting at Line 3: "The FireResp model predicts soil autotrophic and heterotrophic respiration parameterized with a novel dataset across a fire chronosequence in the Yukon and Northwest Territories of Canada. The dataset consisted of soil incubation experiments and field measurements of soil respiration and soil carbon stocks.  The FireResp model contains submodels that consider a $Q_{10}$ (exponential) model of respiration to models of heterotrophic respiration using Michaelis-Menten kinetics parameterized with soil microbe carbon."*

*Additionally we mention the FireResp model in line 61 of the introduction*

Below the main issues that need to be addressed.
2.1)   Line 20 : remove brackets "(2009)".

*We have fixed this citation in line 20*

2.2)   Line 60 : The way the hypothesis is framed confused me during the first reading. What means with "the association of autotrophic respiration with the time … "? In line 253, the association is characterised as positive and I suggest doing so here.

*To clarify this hypothesis more we have split this into two sentences at line 63: "Autotrophic respiration is positively associated with the time since disturbance. This*

*positive association is caused by an underlying positive association of $R_A$ with foliage biomass."*

2.3)   Line 63 : Rs is not defined before. I assume it means soil respiration ?

*At line 21 we have added: "Soil respiration (denoted here as $R_S$) .. "*

2.4)   Figure 1 : the used of colour either shadings or dots, makes the map difficult to read. Forinstance, the colour blue and green used to represent sites 1990 and 1968, are in the same tint as the forest and river of the map. Moreover the layering of shadings and dots makes the different dots hard to distinguish. I suggest using shape instead of colour for dots. There is another red area at the north of the middle map. Is it another fire place ?Finally, the legend is too small.

*Thank you for your suggestions on improving the map.  We have revised the map by (1) choosing a color palette that does not include green (and is also color-blind friendly), (2) using different symbols to represent the different study sites (with the fill color-coded the same as the fire year), and (3) increasing the size of the legend, and (4) including an annotation in the legend "Fire boundary areas are determined from geographic data from the Canadian Wildfire Information System (Natural Resources Canada). The middle inset map also shows additional fire areas burned in 1968 and 2012."*

2.5)   Line 77 : "with a stand replacing fire" - I did not understand what this means.

*At line 80 we have revised this text to "Chronosequence sites were selected from the time since the last fire (in 1968, 1990, and 2012) that burned all aboveground vegetation. We also included a control site, where the last fire was more than 100 years ago."*

*We group the following two comments together:*
2.6)   Lines 92-93 : I am not sure to see how this was computed since incubation data is given in 5 or 10 cm depth. I suggest detailing this step a bit more.
2.7)   Line 94 : It would be interesting to have reference here to support this assumption.

*We have revised this paragraph (lines 94 - 106) to*

*"The field data measured total soil carbon in the top 30 cm, whereas the incubation data included measurements of soil carbon to a given depth (which extended to 50 cm). To determine the total soil carbon to a given depth in the field data we applied a multistep process. This process assumes that the soil carbon profiles in the incubation and field data are similar. First, for the soil carbon in each of the incubation samples (for each replicate line and plot described above) we computed the cumulative proportion of soil carbon (g C m$^2$) to 50 cm (dots in Figure 2). We acknowledge that soil carbon is present in deeper layers (estimated to be 59100 g C m$^2$in the top 100 cm*

*at our sites, see Hugelius et al. (2013) and [https://bolin.su.se/data/ncscd/](https://bolin.su.se/data/ncscd/)). However the objective of this process is a representative empirical estimate of soil carbon for the field data. … "*

2.8)   Figure 2 : Where does the point (1,50) comes from ? Same as the figure 1, it is hard to distinguish colour of the dots.

*We have included the different lines (represented by shapes) and revised the caption to "The points in each plot represents a measurement from an incubation sample, determined from three different lines (represented by different shapes) at each chronosequence site, and within each line, three replicate plots (represented by different colors, Koster et al. (2017)"*

2.9)   Table 1 : It would make the reading of the table easier if the acronyms are defined in the
Legend.

*Based on similar comments from both reviewers we have revised this table to define all the parameters in the legend, removing the need for superscripts on the column headers.  We also noticed that we had erroneously swapped the column labels for $T_{soil}$ and $R_H$ (but not the values).  We apologize for this error.*

2.10)   Line 101 : First occurrence of the FireResp model. It was not introduced before.

*At line 59 we have added: "For this study we synthesize both types of measurements across the chronosequence to parameterize a process model of $R_A$ and $R_H$ (German et al., 2012; Todd-Brown et al., 2012; Sihi et al., 2016), which we call the FireResp model. The FireResp model contains submodels that represent a continuum of complexity in modeling soil carbon. "*

2.11)   Line 112 : "yield" and "assume".

*Thank you for this correction - we have made this change at lines 120 and 121.*

2.12)   Line 123 : The equation 1 includes the volumetric soil moisture fw, after simplifying the
equation through defining gx, fw is not used in gx nor Rx.

*We have corrected this so that $g_X$ includes $f_W$ in line 123- thank you!  We also note that we revised the scaling by fW to a functional form from Moyano et al (2013) as suggested by Reviewer 1 (see comment 1.8)*

2.13)   Line 139 : the end of the sentence is a bit confusing.

*We have removed this phrase and revised line 145 to read: "The Null submodel assumes soil carbon consists of a single pool (Davidson et al., 1998; Reichstein and Beer, 2008)."*

2.14)   Line 177 : This paragraph is not clear to me. I don't get why examining the effect of incubation data on the model is important. Why is it a good thing to reduce the estimation from field data ? Was it an objective of the paper ?

*We have revised this line so that it should be more clear at line 178: "For parameter estimation, we applied a quasi-factorial design with the field and incubation data. This design allowed us to investigate how predictions for $R_A$ and $R_H$ varied when different data are incorporated into the parameter estimation routine. Four different data combinations were used for parameter estimation:"*

2.15)   Line 178: even though it was defined before, I suggest specifying here again what RH refers to.

*Good idea.  At line 187 we have revised this paragraph to "For parameter estimation, we applied a quasi-factorial design with the field and incubation data. This design allowed us to investigate how predictions for autotrophic ($R_A$) and heterotrophic ($R_H$) respiration varied when different data are incorporated into the parameter estimation routine... "*

*We group the following two comments together:*
2.16)   Figure 3: in the caption, "Rows refer to the chronosequence site".
2.17)   Figure 4 : same as figure 3 "Rows refer to ..."

*Thank you for this catch - we have removed both phrases from this revision - this was from an earlier (draft) version of these figures.*

2.18)   Line 264 : "We attributed the difference between figures 3 and 4 to the range ... "

*Now revised at line 273  "We attributed the differences between Figures 3 and Figure 4 is that the soil temperatures from the incubation experiments spanned 1 - 19 $^o$C, allowing for a wider temperature range to characterize any exponential temperature profile..."*

2.19)   Line 326 : How "overwhelmingly edge hitting" is defined ? As far as I understand, we are talking about the site 1990 and the computation of RA. Parameters associated are then Q10,R, kR or gR. And from what I see in figure 5 and supplementary Information, gR and Q10,R are edge hitting for all submodels. How the no impact on the near-zero values of RA was evaluated ?

*We have revised this in Line 338 to: "(Otherwise the values for these aforementioned parameters in Figure 5 or the Supplementary Information for these parameters for all the different models and approaches would be edge-hitting and indicated with the blue colored dots.)"*